# Risk of Cervical Carcinoma After Unfavorable Behavior and High Genetic Risk in the UK Biobank: A Prospective Nested Case–Control Study

**DOI:** 10.3390/biomedicines13020464

**Published:** 2025-02-13

**Authors:** Shiyi Liu, Yunlong Guan, Shitong Lin, Peng Wu, Qing Zhang, Tian Chu, Ruifen Dong

**Affiliations:** 1Department of Gynecology and Obstetrics, Union Hospital, Tongji Medical College, Huazhong University of Science and Technology, Wuhan 430000, China; shiyiliu@whu.edu.cn (S.L.); 2024XH8020@hust.edu.cn (S.L.); pengwu8626@tjh.tjmu.edu.cn (P.W.); 2National Clinical Research Center for Gynecology and Obstetrics, Tongji Hospital, Tongji Medical College, Huazhong University of Science and Technology, Wuhan 430000, China; 3Cancer Biology Research Center (Key Laboratory of the Ministry of Education), Tongji Hospital, Tongji Medical College, Huazhong University of Science and Technology, Wuhan 430000, China; 4Department of Epidemiology and Biostatistics, School of Public Health, Tongji Medical College, Huazhong University of Science and Technology, Wuhan 430000, China; guanyl@hust.edu.cn; 5Department of Obstetrics and Gynecology, Qilu Hospital of Shandong University, Jinan 250000, China; qiluqingzhang@sdu.edu.cn; 6Gynecology Oncology Key Laboratory, Qilu Hospital of Shandong University, Jinan 250000, China; 7Department of Gynecologic Oncology, Tongji Hospital, Tongji Medical College, Huazhong University of Science and Technology, Wuhan 430000, China

**Keywords:** cervical carcinoma, polygenic risk score, PRS, prospective nested case–control study, sexual and reproductive behavior, public health, risk factors

## Abstract

**Background**: Previous studies have established a general understanding of the association between risky sexual behavior, genetic risk, and cervical carcinoma. However, these studies were conducted several years ago and lack systematic analysis using high-quality and population-based data. **Methods**: We conducted a prospective nested case–control study to identify risky behaviors and developed a behavior score. Combining the behavior score and genetic risk, we evaluated the effect of sexual and reproductive behavior and PRS on cervical carcinoma through the developed conditional logistic regression models. **Results**: We verified increased carcinoma risk in individuals with early sexual intercourse (OR: 1.41 [95% CI 1.09 to 1.83], *p =* 0.0083), non-monogamous sexual partners (OR: 3.13 [95% CI 2.15 to 4.57], *p* < 0.0001), three or more live births (OR: 1.44 [95% CI 1.12 to 1.84], *p =* 0.0040), and high PRS (polygenic risk score) (top 25% of PRS, OR: 1.58 [95% CI 1.15 to 2.16], *p =* 0.0044). The unfavorable sexual and reproductive behavior score we developed was linked to a 151% increased risk (OR: 2.51 [95% CI 1.79 to 3.52], *p* < 0.0001) after adjusting for PRS. Women with both unfavorable behavior and high genetic risk had a 5.5-fold increased cervical carcinoma risk (OR: 5.45 [95% CI 2.72 to 10.95], *p* < 0.0001) compared to individuals with favorable behavior and low genetic risk. **Conclusions**: Unfavorable sexual and reproductive behavior increases the risk of cervical carcinoma, especially in those with a high genetic risk. These findings encourage us to adhere to a healthy sexual and reproductive pattern.

## 1. Introduction

Cervical carcinoma stands as the foremost cause of cancer-related deaths among women across the globe, recording an incidence rate of 13.3 cases per 100,000 woman-years in 2020 [1]. In response, the World Health Organization (WHO) has initiated an ambitious endeavor aimed at eradicating cervical carcinoma by 2030, emphasizing the paramount importance of addressing this malady on a global public health level [2]. In low-income and middle-income countries, such as Eastern Africa, where effective screening and HPV vaccination initiatives remain scarce, women bear the brunt of inadequate primary prevention and early detection measures [3,4,5]. In these regions, the incidence rate is at least three times higher compared to more affluent areas, and there has even been a surge in cases in recent years [4,6,7,8]. Conversely, high-income nations encounter their own set of challenges; although the incidence rate seems to have reached a plateau [9], it has not demonstrated a decline, thereby exposing a substantial disparity with the WHO’s objective: reducing the incidence to 4 cases per 100,000 woman-years by the year 2030 [2].

Persistent high-risk human papillomavirus (hrHPV) infection stands as the primary underlying cause of cervical carcinoma and is mainly transmitted through sexual contact, including vaginal, anal, and oral sex [10,11,12]. There is some evidence that risky sexual behaviors, such as engaging with multiple male partners and initiating sexual activity at a young age, along with experiencing multiple full-term pregnancies, contribute to an increased vulnerability to hrHPV infection and consequently raise the risk of cervical carcinoma [13]. Therefore, fostering a commitment to wholesome sexual and reproductive behavior, facilitated by patient awareness and comprehensive educational initiatives, should be an essential component of efforts aimed at eradicating cervical carcinoma.

Beyond the influence of environmental factors, the hereditary/genetic component also plays a substantial role in determining the vulnerability to cervical carcinoma [14]. The genetic contribution is estimated to account for a substantial portion, around 27–36%, of the total risk associated with the development of cervical carcinoma [15]. The genetic risk loci identified in genome-wide association studies (GWASs) have proven useful in devising a polygenic risk score (PRS), enabling the estimation of an individual’s inherent genetic predisposition to conditions like preeclampsia and gestational hypertension [16]. Regrettably, the field of cervical carcinoma research has yet to witness the application of a PRS based on these identified risk variants aimed at evaluating the likelihood of developing cervical carcinoma.

In this study, we hypothesized that sexual and reproductive behavior is associated with cervical carcinoma risk. By integrating this insight with genetic risk, our aim was to devise a cost-effective approach that can offer recommendations for cervical carcinoma prevention on a global level.

## 2. Methods

### 2.1. Study Design and Participants

We conducted a prospective matched nested case–control study using data from female participants of European ancestry in the UK Biobank cohort [17]. The UK Biobank is a large prospective cohort study that encompasses over 500,000 individuals aged from 40 to 69, residing within communities across the United Kingdom from 2006 to 2010. These participants congregated at 22 assessment centers distributed throughout England, Scotland, and Wales. At these centers, they diligently fulfilled surveys and underwent comprehensive physical measurements, collectively contributing to the wealth of data amassed by the study. We obtained data on 21 July 2022 and set that date as the end-of-follow-up date.

We excluded women diagnosed with either cervical carcinoma in situ or invasive cervical carcinoma (D06 and C53) prior to enrollment as determined by the predefined International Classification of Diseases codes (ICD10). Additionally, we excluded women who had received a diagnosis of carcinoma other than in situ or invasive cervical carcinoma by 21 July 2022. Further exclusion criteria included (1) age at first sexual intercourse being either unknown or less than 12 years or women with no previous sexual intercourse; (2) unknown number of sexual partners; (3) unknown number of live births; and (4) unknown smoking status. The application number of the UK Biobank in our study was 82,232.

### 2.2. Procedures and Outcomes

Incident cervical carcinoma cases were matched to controls in a 1:20 ratio based on age, smoking status, and year of enrollment using time-dependent incidence density sampling. Because smoking cigarettes can elevate the risk of cervical carcinoma by twofold [18,19], we took careful measures to align smoking status during the cohort development. By employing this method, an equivalent duration of follow-up was allotted to both cases and their corresponding controls, thereby guaranteeing uniform exposure periods for accurate comparisons.

The main outcome was cervical carcinoma, and exposures of interest were three sexual and reproductive behaviors and genetic risk. Cervical carcinoma was defined as carcinoma in situ or invasive carcinoma of the uterine carcinoma. Sexual and reproductive behaviors included age at first sexual intercourse, number of male sexual partners, and number of live births (see Appendix A).

Risky sexual and reproductive behaviors included first sexual intercourse at an age earlier than 18 years, non-monogamous sexual partners, and three or more live births. We defined sexual and reproductive scores based on the number of risk factors, so the scores ranged from 0 to 3. Sexual and reproductive behavior was classified as “favorable” (0 or 1 risk factor), “intermediate” (2 risk factors), or “unfavorable” (3 risk factors). The genetic risk was determined by the individual’s PRS and categorized into three groups: “low genetic risk” (bottom 25% of PRS), “intermediate genetic risk” (25th to 75th percentile), and “high genetic risk” (top 25% of PRS).

### 2.3. Polygenic Risk Score

We constructed the PRS (see Appendix A, Appendix A) for cervical carcinoma for individuals in the UK Biobank using 21 significant and independent variants based on the cervical carcinoma GWAS in the FinnGen study (data freeze 8, Fall 2021, Appendix A) [20].

### 2.4. Statistical Analysis

Univariable and multivariable conditional logistic regressions were used to assess associations between the sexual and reproductive behavior and PRS with cervical carcinoma status. Odds ratios (ORs) with 95% confidence intervals (CIs) were calculated to determine the odds of developing cervical carcinoma when comparing participants with a given exposure to those without.

First, unadjusted ORs with 95% CIs were calculated by comparing cases and controls (model 1). Model 2 adjusted for body mass index (BMI) [21] and oral contraceptive use to account for residual confounding. Model 3 further incorporated PRS in addition to covariates in model 2 as well as the first ten principal components of genetic ancestry and genotyping batch.

In the sensitivity analyses, we further refined the risk ratios of different sexual and reproductive behaviors and assigned different score weights based on their regression coefficients [22]. Therefore, sexual and reproductive behavior scores ranged from 0 to 6 and were further classified as “favorable” (score 0 to 2), “intermediate” (score 3 to 5), or “unfavorable” (score 6). Apart from performing the sensitivity analysis by assigning different score weights, we also performed two additional nested case–control matches with matching ratios of 1:10 and 1:40.

All tests were two-sided with statistical significance at *p* less than 0.05. We also performed power calculations prior to the study (Appendix A). All statistical analyses were performed using R software version 4.3.1 (packages “Epi”, “epiR”, and “survival”).

## 3. Results

The final analysis involved 7035 participants from the UK Biobank (Figure 1). Table 1 provides an overview of the baseline characteristics of the study population. Within the nested case–control study, there were 335 women with incident cervical carcinoma and 6700 matched controls. When comparing cases with in situ or invasive cervical carcinoma to controls, several notable differences emerged. Cases had a younger age at first sexual intercourse (234 of 335 [69.85%] vs. 4235 of 6700 [63.21%]), a higher number of sexual partners (non-monogamous partners: 302 of 335 [90.15%] vs. 5123 of 6700 [76.46%]), and a greater number of live births (greater than two live births: 95 of 335 [28.36%] vs. 1456 of 6700 [21.73%]).

Our constructed PRS could well stratify the risk of cervical carcinoma among the female individuals of European ancestry in the UK Biobank (see Appendix A). In the conditional logistic regression analysis for matched cases and controls, individuals with high PRS had a more pronounced elevated risk (OR: 1.58 [95% CI 1.15 to 2.16], *p =* 0.0044; Table 2) compared to those with a low PRS after adjusting for BMI and oral contraceptives.

In the unadjusted model, participants who reported early first intercourse had a higher risk of cervical carcinoma (OR: 1.41 [95% CI 1.09 to 1.82], *p =* 0.0082; Table 3). Similarly, engaging in non-monogamous partners (OR: 3.13 [95% CI 2.14 to 4.56], *p* < 0.0001) and having multiple live births (OR: 1.43 [95% CI 1.12 to 1.84], *p =* 0.0042) were also associated with an increased incidence of cervical carcinoma. Even after adjusting for relevant covariates, the aforementioned risky sexual behaviors remained significantly associated with cervical carcinoma incidence (age at first sexual intercourse before 18: OR: 1.41 [95% CI 1.09 to 1.83], *p =* 0.0083; non-monogamous sexual partners: OR: 3.13 [95% CI 2.15 to 4.57], *p* < 0.0001; three or more live births: (OR: 1.44 [95% CI 1.12 to 1.84], *p =* 0.0040; Table 3). Furthermore, when incorporating all exposures and covariates of interest into the regression model, individuals with non-monogamous sexual partners (OR: 3.07 [95% CI 2.09 to 4.50], *p* < 0.0001; Table 4) and three or more live births (OR: 1.48 [95% CI 1.15 to 1.90], *p =* 0.0023) continued to exhibit a high risk of cervical carcinoma. After further introducing genetic risk in the model, sexual and reproductive risk factors remained statistically significant (non-monogamous sexual partner: OR: 3.04 [95% CI 2.07 to 4.47], *p* < 0.0001; three or more live births: OR: 1.47 [95% CI 1.15 to 1.90], *p =* 0.0024; high PRS: (OR: 1.60 [95% CI 1.17 to 2.19], *p =* 0.0035; Table 5).

We formulated a sexual and reproductive behavior score for each individual by amalgamating and categorizing exposure factors from the aforementioned studies. The risk of cervical carcinoma significantly increased for individuals with unfavorable behavior (unadjusted OR: 2.55 [95% CI 1.83 to 3.57], *p* < 0.0001; Table 6; adjusted OR: 2.51 [95% CI 1.79 to 3.52], *p* < 0.0001; Table 7) and those with intermediate behavior (unadjusted OR: 1.64 [95% CI 1.25 to 2.15], *p* = 0.00033; adjusted OR: 1.64 [95% CI 1.24 to 2.15], *p =* 0.00042) when compared to individuals with favorable sexual and reproductive behavior. After considering genetic risk, participants with unfavorable sexual and reproductive behavior remained at a heightened risk of cervical carcinoma (adjusted OR: 2.51 [95% CI 1.79 to 3.52], *p* < 0.0001). There was no interaction between sexual and reproductive behavior and genetic risk (Table 8).

Table 9 displayed the crude risk of cervical carcinoma concerning combined genetic risk and sexual and reproductive behavior. Participants with unfavorable behavior and a high genetic risk exhibited an exceptionally elevated risk of developing cervical carcinoma (adjusted OR: 5.45 [95% CI 2.72 to 10.95], *p* < 0.0001; Figure 2) after adjusting for covariates and PRS. Furthermore, after stratification according to genetic risk, participants in the low PRS group had an increased risk of developing the disease as a result of an unfavorable behavior (unfavorable sexual and reproductive behavior in the low PRS group: OR: 2.89 [95% CI 1.23 to 6.8], *p =* 0.015; Table 10).

In the sensitivity analyses, an unfavorable behavior showed a significant increase in cervical carcinoma risk (adjusted OR: 4.00 [95% CI 2.58 to 6.2], *p* < 0.0001; Appendix A). Participants with an unfavorable behavior and a high genetic risk had significantly increased risk ratios (adjusted OR: 8.90 [95% CI 3.47 to 22.79], *p* < 0.0001; Appendix A). Meanwhile, our results remained robust after additional matching based on different ratios. Participants with unfavorable behavior and a high genetic risk were still exposed to extremely high cervical carcinoma risks (adjusted OR: 4.5 [95% CI 2.2 to 9.22], *p* < 0.0001 at matching ratio 1:10, Appendix A, Appendix A; adjusted OR: 5.28 [95% CI 2.66 to 10.47], *p* < 0.0001 at matching ratio 1:40, Appendix A, Appendix A).

## 4. Discussion

Using a prospective nested case–control design, the current study delved into the joint effect of sexual and reproductive behavior and genetic risk on the incidence of cervical carcinoma. In the course of our investigation, based on a population of over 7000 individuals drawn from the UK Biobank cohort, we identified several noteworthy independent risk factors. It became evident that the presence of non-monogamous male sexual partners served as a robust marker for the incident evaluation of cervical carcinoma. This trend was paralleled by two additional factors: initiating heterosexual intercourse before the age of 18 and undergoing three or more childbirths. It is crucial to emphasize that these associations remained significant independently of genetic risk. A study conducted by Sarah J. Bowden et al. [15] identified an increased risk of cervical carcinoma associated with having multiple partners. Moreover, 54% of patients with carcinoma and 26% of controls reported extramarital partners [23]. Engaging in multiple partnerships amplifies the probability of contracting hrHPV, with each additional partner further heightening the potential for exposure to the virus [24]. The observed correlation between younger age at first sexual intercourse and an augmented risk of cervical carcinoma aligns with findings from prior research, but the precise mechanisms remain to be elucidated [25]. Younger females may have an immature cervix that is more susceptible to HPV infection, and early sexual debut often correlates with a higher number of sexual partners, increasing the likelihood of exposure to high-risk HPV strains. The biological vulnerability of the cervix in early sexual initiation may also be compounded by an underdeveloped immune system, leading to inadequate clearance of HPV infections [26].

Previous studies surrounding the association between pregnancy and cervical carcinoma have not yet yielded a definitive outcome [23]. Some studies have suggested that higher parity increases the risk of cervical carcinoma [27], while others have found no discernible connection between pregnancy and the incidence of cervical carcinoma [23,28].

In our current study, we contribute fresh insights that shed light on this matter, revealing that having three or more live births indeed heightens the risk of cervical carcinoma. However, the precise reasons behind this association remain somewhat unclear. Previous investigations have not furnished definitive evidence supporting a higher prevalence of hrHPV among pregnant women when compared to non-pregnant individuals [28]. An alternative hypothesis suggests that the elevation in estrogen and progesterone levels during pregnancy could trigger cervical ectopy [29,30], thereby potentially facilitating direct exposure to HPV and other cofactors.

These three behavioral factors form a critical foundation for the development of sexual and reproductive behavior scores in the context of cervical carcinoma. As the number of risk behaviors increases, carcinoma risk emerges. Unfavorable sexual and reproductive behavior was linked to a twofold increase in the risk of incidence when compared to favorable behavior. The significance of uncovering modifiable factors as part of cervical carcinoma prevention strategies cannot be overstated, given the potential for reducing risks within the female population.

As for genetic risk, PRS is an evaluation tool that is applied in several diseases, including ovarian cancer [31], endometrial cancer [32], and uterine leiomyoma [33]. Our previous study based on GWAS attempted to explore the genetic variations of cervical carcinoma and identified novel variants rs13117307 and rs8067378 that are linked to an increased risk of cervical cancer [34]. Nonetheless, the utilization of PRS based on genetic variants to evaluate the susceptibility to cervical carcinoma remains an area with limited research. Our findings highlighted a clear connection between high PRS and an elevated likelihood of cervical carcinoma, independent of sexual and reproductive behavior. By combining the sexual and reproductive behavior with the PRS, we effectively stratified females into distinct risk categories. Notably, individuals characterized by unfavorable sexual and reproductive behavior coupled with a high PRS exhibited a more than fourfold increase in susceptibility to cervical carcinoma compared to those with favorable behavior and low genetic risk. Additionally, the risk mitigation attributed to adopting a healthy sexual and reproductive behavior was consistent across all genetic risk levels. This finding emphasizes the universal benefits of adhering to secure and healthy sexual and reproductive behavior. Moreover, even those with lower polygenic risk profiles can face a significantly amplified risk of developing cervical carcinoma if they engage in unfavorable sexual and reproductive behavior. This highlights the vital importance of maintaining safe and healthy sexual and reproductive behavior for everyone.

The potential clinical applications stemming from this study can be broadly categorized into primary and secondary prevention strategies. Our findings indicate that the GWAS-derived PRS shows potential as an early marker of cervical carcinoma risk. This is particularly significant as it might be used even before HPV infection testing as an evaluation tool. This underscores the feasibility of utilizing PRS to guide targeted health education and interventions. In practical terms, PRS could help pinpoint individuals, especially those with a high genetic risk, who might benefit from tailored strategies to encourage safer sexual and reproductive behavior and HPV vaccination. Such interventions could play a crucial role in mitigating the risk of cervical carcinoma in these high-risk demographics.

Cervical carcinoma screening has remained at the forefront of the battle against the disease for over six decades [35]. The current recommendations from the U.S. Preventive Services Task Force (USPSTF) recognize special populations, including HIV-positive females and those with compromised immune systems [36]. For HIV-positive females, it is advised that subsequent screenings are conducted annually after the initial cervical cytology screening. However, the importance of considering sexual and reproductive behavior for screening recommendations has not been sufficiently emphasized in the USPSTF guidelines. This study has the potential to complement the existing USPSTF recommendations for cervical carcinoma screening. By leveraging the combined sexual and reproductive behavior with PRS, it is feasible to identify high-risk individuals who might benefit from earlier or more frequent screenings. This approach aligns with the broader goal of adjusting the interval between cervical carcinoma screenings based on individual risk factors, which, in turn, leads to improved detection of high-grade cervical precancer. The WHO has pointed out that low-income and middle-income countries face challenges in implementing cytology-based screening programs, often resulting in limited coverage [2]. This study offers a cost-effective strategy that could be particularly beneficial for low-income and middle-income countries with constrained resources. By targeting females who encounter challenges such as early marriage, multiple sexual partners, and having multiple children, these regions could make optimal use of available resources to ensure equitable access to cervical carcinoma prevention strategies.

### Strengths and Limitations

In this prospective nested case–control study, we recruited a sufficient number of participants to evaluate the risk of cervical carcinoma by the combination of sexual and productive behavior scores and PRS. Numerous factors were carefully matched to minimize selection bias [37]. However, our study has a few limitations. Firstly, our analytical scope was confined to individuals of European ancestry, with an average age of 44, who did not have a history of HPV vaccination. This emphasizes the need for future research to extend these investigations to encompass a more diverse and comprehensive demographic landscape. Secondly, our analysis specifically focused on a limited set of sexual-related factors. Expanding the scope of inquiry to include additional factors, such as the duration of exposure to certain risk elements like sexual frequency, could provide a more comprehensive understanding of the relationships at play. Despite these limitations, our analysis remains significant as it stands as the largest and inaugural examination to date into the influence of both modifiable factors and unmodifiable factors on the incidence of cervical carcinoma.

## 5. Conclusions

This prospective nested case–control study identified several noteworthy independent risk factors of cervical carcinoma, including non-monogamous male sexual partners, initiating heterosexual intercourse before the age of 18, undergoing three or more childbirths, and high PRS. The risk of cervical carcinoma increases with the accumulation of these risk factors. This study underscores the critical importance of maintaining safe and healthy sexual and reproductive practices for individuals, regardless of their PRS profiles.

## Figures and Tables

**Figure 1 biomedicines-13-00464-f001:**
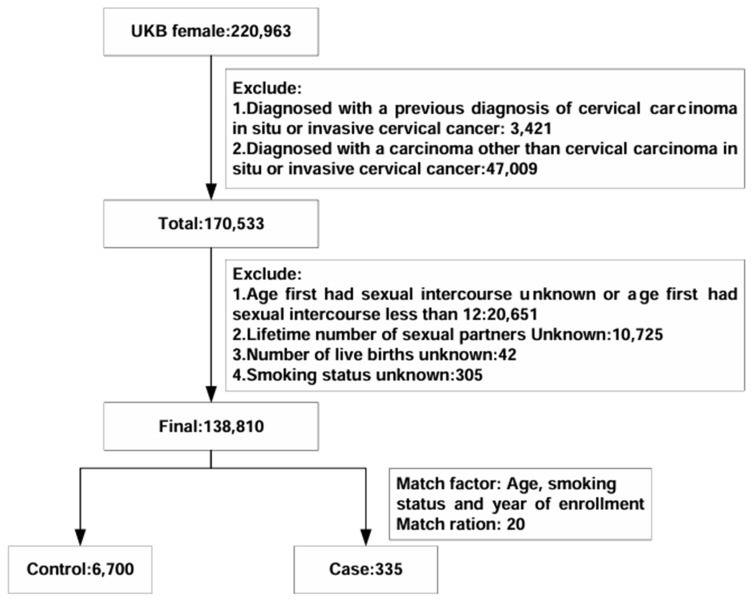
Cohort exclusions and nested case–control design.

**Figure 2 biomedicines-13-00464-f002:**
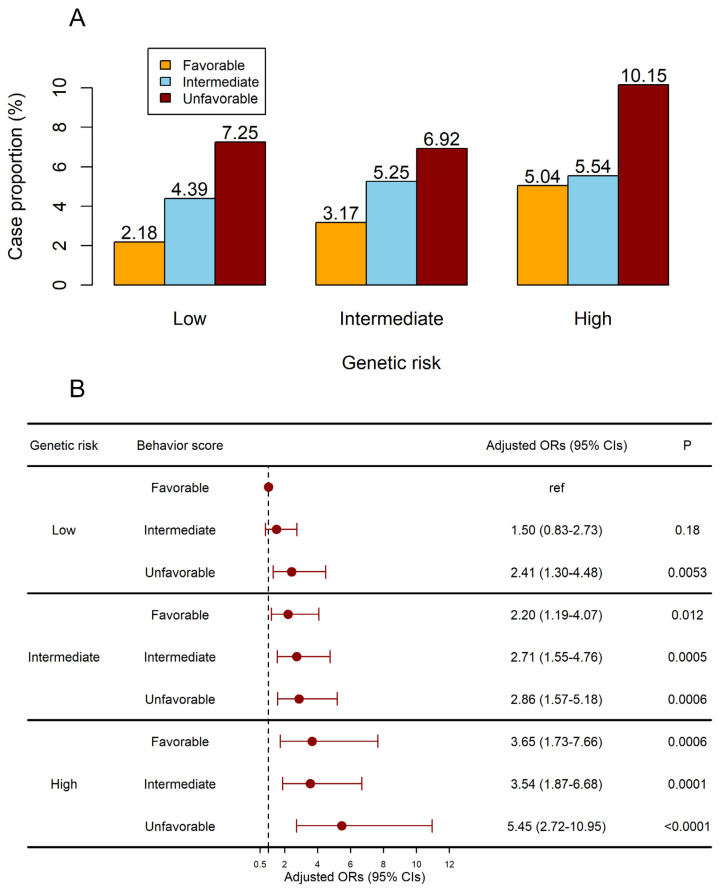
Joint effects of the sexual and reproductive behavior score and PRS on the risk of cervical carcinoma. (**A**) The case proportion in subgroups. (**B**) The adjusted ORs of subgroups. Models were adjusted for BMI and oral contraceptives.

**Table 1 biomedicines-13-00464-t001:** Characteristics of the study population with cervical carcinoma cases and controls matched by age, sex, smoking status, and year of enrollment, UK Biobank 2006–2010.

Characteristics		Overall	Control	Case
		7035	6700	335
Age at first intercourse (%)	≤18	4469 (63.53)	4235 (63.21)	234 (69.85)
	>18	2566 (36.47)	2465 (36.79)	101 (30.15)
Number of sexual partners (%)	1	1610 (22.89)	1577 (23.54)	33 (9.85)
	≥2	5425 (77.11)	5123 (76.46)	302 (90.15)
Live births (%)	≤2	5484 (77.95)	5244 (78.27)	240 (71.64)
	≥3	1551 (22.05)	1456 (21.73)	95 (28.36)
Genetic risk (%)	Low	1760 (25.02)	1692 (25.25)	68 (20.30)
	Intermediate	3519 (50.02)	3355 (50.07)	164 (48.96)
	High	1756 (24.96)	1653 (24.67)	103 (30.75)
Base age (median [IQR])		50.00 [44.00, 56.00]	50.00 [44.00, 56.00]	50.00 [44.00, 56.00]
Townsend deprivation index (median [IQR])		−2.1 [−3.6, 0.4]	−2.1 [−3.6, 0.4]	−1.9 [−3.5, 0.6]
Smoking status (%)	Current	1071 (15.22)	1020 (15.22)	51 (15.22)
	Never	3885 (55.22)	3700 (55.22)	185 (55.22)
	Previous	2079 (29.55)	1980 (29.55)	99 (29.55)
Year of enrollment (%)	2007	1029 (14.63)	980 (14.63)	49 (14.63)
	2008	2877 (40.90)	2740 (40.90)	137 (40.90)
	2009	2121 (30.15)	2020 (30.15)	101 (30.15)
	2010	1008 (14.33)	960 (14.33)	48 (14.33)

**Table 2 biomedicines-13-00464-t002:** Association between cervical carcinoma and genetic risk in a nested case–control study.

**Characteristics**		**Crude OR ^a^ (95% CI)**	** *p* **	**Adjusted OR ^b^ (95% CI)**	*p*
Genetic risk category	Low	1		1	
	Intermediate	1.24 (0.93–1.66)	0.1403	1.25 (0.94–1.67)	0.1283
	High	1.58 (1.15–2.16)	0.0044	1.58 (1.16–2.17)	0.0042

^a^: Calculated using a multivariable conditional logistic regression model, adjusted for the first ten principal components of genetic ancestry and genotyping batch. ^b^: Calculated using a multivariable conditional logistic regression model, adjusted for the first ten principal components of genetic ancestry, genotyping batch, BMI, the Townsend deprivation index, and oral contraceptive use.

**Table 3 biomedicines-13-00464-t003:** Association between cervical carcinoma and behavioral factors in a nested case–control study.

Characteristics		Crude OR ^a^ (95% CI)	*p*	Adjusted OR ^b^ (95% CI)	*p*
Age at first intercourse	>18	1		1	
	≤18	1.41 (1.09–1.82)	0.0082	1.41 (1.09–1.83)	0.0087
Number of sexual partners	1	1		1	
	≥2	3.13 (2.14–4.56)	<0.0001	3.09 (2.12–4.52)	<0.0001
Live births	≤2	1		1	
	≥3	1.43 (1.12–1.84)	0.0042	1.43 (1.12–1.83)	0.0047

^a^: Calculated using a univariable conditional logistic regression model. ^b^: Calculated using a multivariable conditional logistic regression model, adjusted for BMI, the Townsend deprivation index, and oral contraceptive use.

**Table 4 biomedicines-13-00464-t004:** Multivariable conditional logistic regression of cervical carcinoma risk (Adjusted for behavioral factors, BMI, Townsend index, and oral contraceptive use).

Characteristics		Adjusted OR ^a^ (95% CI)	*p*
Age at first intercourse	>18	1	
	≤18	1.16 (0.89–1.50)	0.2634
Number of sexual partners	1	1	
	≥2	3.04 (2.07–4.46)	<0.0001
Live births	0–2	1	
	≥3	1.47 (1.15–1.89)	0.0025

^a^: Calculated using a multivariable conditional logistic regression model, adjusted for behavioral factors, BMI, the Townsend deprivation index, and oral contraceptive use.

**Table 5 biomedicines-13-00464-t005:** Multivariable conditional logistic regression of cervical carcinoma risk (Adjusted for behavioral factors, BMI, oral contraceptive use, Townsend index, PRS, and genetic ancestry).

Characteristics		Total	Case	Adjusted OR ^a^ (95% CI)	*p*
Age at first intercourse	>18	2558	101 (3.95)	1	
	≤18	4477	234 (5.23)	1.14 (0.88–1.48)	0.3155
Number of sexual partners	1	1578	33 (2.09)	1	
	≥2	5457	302 (5.53)	3.00 (2.04–4.41)	<0.0001
Live births	0–2	5551	240 (4.32)	1	
	≥3	1484	95 (6.4)	1.47 (1.14–1.89)	0.0027
Genetic risk category	Low	1759	71 (4.04)	1	
	Intermediate	3517	162 (4.61)	1.24 (0.93–1.66)	0.1450
	High	1759	102 (5.8)	1.60 (1.17–2.20)	0.0033

^a^: Calculated using a multivariable conditional logistic regression model, adjusted for behavioral factors, BMI, oral contraceptive use, the Townsend deprivation index, PRS, the first ten principal components of genetic ancestry, and genotyping batch.

**Table 6 biomedicines-13-00464-t006:** Univariable conditional logistic regression of cervical carcinoma risk and sexual and reproductive behavior scores.

Characteristics		Crude OR (95% CI)	*p*
Behavior Score	Favorable	1	
	Intermediate	1.64 (1.25–2.15)	0.00033
	Unfavorable	2.55 (1.83–3.57)	<0.0001

**Table 7 biomedicines-13-00464-t007:** Association between cervical carcinoma and sexual and reproductive behavior and genetic risk in a nested case–control study.

**Characteristics**		**Adjusted OR ^a^ (95% CI)**	** *p* **	**Adjusted OR ^b^ (95% CI)**	*p*
Behavior Score	Favorable	1		1	
	Intermediate	1.63 (1.24–2.15)	0.00041	1.62 (1.23–2.13)	0.00053
	Unfavorable	2.53 (1.80–3.55)	<0.0001	2.47 (1.76–3.48)	<0.0001
Genetic risk category	Low			1	
	Intermediate			1.24 (0.93–1.66)	0.14205
	High			1.59 (1.16–2.18)	0.00040

^a^: Calculated using a multivariable conditional logistic regression model, adjusted for BMI, the Townsend deprivation index, and oral contraceptive use. ^b^: Calculated using a multivariable conditional logistic regression model, adjusted for BMI, the Townsend deprivation index, oral contraceptive use, PRS, the first ten principal components of genetic ancestry, and genotyping batch.

**Table 8 biomedicines-13-00464-t008:** Additive interactions between sexual and reproductive behavior and genetic risk on the risk of cervical carcinoma.

Behavior Score	Intermediate Genetic Risk ^a^		High Genetic Risk		*p* for Interaction
	RERI (95% CI)	AP (95% CI)	RERI (95% CI)	AP (95% CI)	
Intermediate	0.01 (−1.13, 1.15)	0.00 (−0.42, 0.43)	−0.76 (−2.40, 0.88)	−0.27 (−0.83, 0.30)	0.415
Unfavorable	−0.55 (−3.01, 1.91)	−0.16 (−0.87, 0.55)	0.38 (−2.87, 3.63)	0.07 (−0.51, 0.66)	

^a^: Models were adjusted for BMI, the Townsend deprivation index, oral contraceptive, the first ten principal components of genetic ancestry, and genotyping batch. RERI, the relative excess risk due to interaction; AP, the attributable proportion due to interaction.

**Table 9 biomedicines-13-00464-t009:** Risk of cervical carcinoma according to genetic and behavior profiles.

Genetic Risk Category	Behavior Score		
	Favorable	Intermediate	Unfavorable
Low	1 (reference)	2.18 (1.18–4.03) *p* = 0.0130	3.56 (1.69–7.49) *p* = 0.0008
Intermediate	1.51 (0.83–2.74) *p* = 0.1777	2.70 (1.54–4.74) *p* = 0.0006	3.52 (1.86–6.65) *p* < 0.0001
High	2.42 (1.30–4.49) *p* = 0.0052	2.83 (1.56–5.14) *p* = 0.0006	5.36 (2.67–10.77) *p* < 0.0001

Calculated using a multivariable conditional logistic regression model, adjusted for BMI, the Townsend deprivation index, oral contraceptive use, the first ten principal components of genetic ancestry, and genotyping batch.

**Table 10 biomedicines-13-00464-t010:** Association of behavior scores with cervical carcinoma incidence according to genetic risk strata.

Characteristics		Low Genetic Risk		Intermediate Genetic Risk		High Genetic Risk	
		Adjusted OR ^a^ (95% CI)	*p*	Adjusted OR ^a^ (95% CI)	*p*	Adjusted OR ^a^ (95% CI)	*p*
Behavior Score	Favorable	1		1		1	
	Intermediate	2.23 (1.10–4.53)	0.0264	1.58 (1.06–2.35)	0.0240	0.99 (0.59–1.68)	0.9754
	Unfavorable	3.12 (1.30–7.47)	0.0108	2.31 (1.40–3.84)	0.0012	1.75 (0.89–3.44)	0.1040

^a^: Calculated using a multivariable conditional logistic regression model, adjusted for BMI, the Townsend deprivation index, and oral contraceptive use.

## Data Availability

The genetic and phenotypic UK Biobank data are available on application to the UK Biobank (www.ukbiobank.ac.uk/, accessed on 1 July 2023). The FinnGen GWAS is available from https://www.finngen.fi/en, accessed on 1 July 2023.

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
