# Peer review of "Risk of Cervical Carcinoma After Unfavorable Behavior and High Genetic Risk in the UK Biobank: A Prospective Nested Case–Control Study"

_biomedicines, 2025, doi:10.3390/biomedicines13020464_

Round 1
Reviewer 1 Report
Comments and Suggestions for Authors
-
The Introduction should provide more details on the issues of infection and development of cervical cancer, including possible routes of infection. HPV is most often transmitted through sexual contact with an infected person. Moreover, contact can be any: vaginal, oral or anal.
-
Why were the groups with and without HPV vaccination not separated? Perhaps the group with early sexual intercourse was largely unvaccinated?
-
Were the groups of men (partners) with and without HPV vaccination separated?
-
Were the groups with and without oncogenic HPV types (16, 18, 31, 33, 35, 39, 45, 51, 52, 56, 58, 59 and 66) separated?
-
The authors should provide more details on the relationship between sexual and reproductive behavior and genetic risk in the Conclusion.
-
How exactly can these results be used as a tool for predicting cervical cancer cases?
Author Response
First of all, we would like to express our sincere thanks to your helpful comments and suggestions on our manuscript, your time and efforts in handling our manuscript are much appreciated. The explanation of the modifications as well as corrections in this revision are listed as follow. The changes that we made have been marked with yellow in the revised manuscript.
Comments 1: The Introduction should provide more details on the issues of infection and development of cervical cancer, including possible routes of infection. HPV is most often transmitted through sexual contact with an infected person. Moreover, contact can be any: vaginal, oral or anal.
Response 1: Thanks for your suggestion. More information on the infection and development of cervical cancer has been added as “Persistent high-risk human papillomavirus (hrHPV) infection stands as the primary underlying cause of cervical carcinoma, and is mainly transmitted through sexual contact, including vaginal, anal, and oral sex”. This has been added in the introduction. (“Introduction”, Page 4, line 68-70)
Comments 2: Why were the groups with and without HPV vaccination not separated? Perhaps the group with early sexual intercourse was largely non-vaccinated?
Response 2: Thank you for your question. We were unable to separate the population into vaccinated and non-vaccinated groups in this study because it is based on the UK Biobank, where participants are aged 40 to 69, and their first sexual intercourse occurred before the HPV vaccine became widely available. Most participants did not receive the HPV vaccine during adolescence, as it was not widely available at that time. Additionally, the sample size of the vaccinated population is likely insufficient to support subgroup comparisons. Therefore, we were unable to analyze the vaccinated and non-vaccinated groups separately.
Comments 3: Were the groups of men (partners) with and without HPV vaccination separated?
Response 3: Thank you for your question. We were unable to perform subgroup analyses for men (partners) with and without HPV vaccination, as the UK Biobank does not contain specific data on the HPV vaccination status of partners.
Comments 4. Were the groups with and without oncogenic HPV types (16, 18, 31, 33, 35, 39, 4551.52.56.58.59 and 66) separated?
Response 4: Thank you for your question. The reason for not separating groups based on oncogenic HPV types is the same as for Comment 3. Only a very small proportion of the UK Biobank population has data on HPV infection and vaccination status, and specific information on HPV types is not available. Therefore, conducting a separate analysis for participants with or without oncogenic HPV infections was not feasible in this study. Your suggestion is meaningful, and we hope to solve this limitation in future research by collecting and analyze relevant data (include HPV types) from our own hospital.
Comments 5. The authors should provide more details on the relationship between sexual and reproductive behavior and genetic risk in the Conclusion.
Response 5: Thanks for your suggestion, we have revised conclusion as follow “This prospective nested case-control study identified several noteworthy independent risky factors of cervical carcinoma, including non-monogamous male sexual partner, initiating heterosexual intercourse before the age of 18, undergoing three or more childbirths and high PRS, The risk of cervical carcinoma increases with the accumulation of these risk factors. This study underscores the critical importance of maintaining safe and healthy sexual and reproductive practices for individuals, regardless of their PRS profiles”. (“Conclusion”, Page 16, line 330-336)
Comments 6: How exactly can these results be used as a tool for predicting cervical cancer cases?
Response 6: Thank you for your question. Our GWAS results are not designed for precise individual-level risk prediction. Instead, they provide insights that can be used in public health education, helping to raise awareness of modifiable risk factors and supporting primary prevention strategies. We have also revised the manuscript to replace any references to "predict" with "evaluate" to reflect this distinction more accurately.
Reviewer 2 Report
Comments and Suggestions for Authors
This manuscript presents a prospective nested case-control study exploring the interplay between unfavorable sexual and reproductive behavior, polygenic risk score (PRS), and cervical carcinoma incidence using data from the UK Biobank. The study is relevant and timely, addressing critical public health concerns regarding cervical cancer prevention and risk stratification. However, significant issues related to methodological clarity, data presentation, and contextualization must be addressed before the manuscript is suitable for publication.
Major comments:
1. The construction of the PRS is inadequately described. Simply stating the inclusion of 21 genetic variants is insufficient. Include detailed information on the genetic variants selected, the rationale behind their inclusion, and the validation of the PRS. The manuscript does not address potential confounding factors, such as population stratification or genetic ancestry differences, which may bias the PRS outcomes. Adjustments for these factors should be explicitly stated. The biological relevance of the selected variants. Are these variants causally linked to cervical carcinoma, or are they simply associated?
2. The matching process based on age and smoking status is reasonable, but more justification is needed for these choices over other potentially confounding variables, such as socioeconomic status or HPV vaccination history.
3. Oral contraceptive use is included as an adjustment variable, but its potential as an independent modifier of risk is ignored. This oversight undermines the interpretability of the results.
4. Interaction effects between PRS and behavioral scores are only briefly mentioned but not rigorously tested. A detailed assessment using interaction terms or stratified analyses is necessary to validate the combined risk model.
5. Why does high PRS contribute relatively modestly (OR 1.58) to risk compared to behavioral factors (e.g., OR 3.13 for multiple partners)? This requires deeper exploration of the biological underpinnings.
6. How do the authors reconcile the overlapping confidence intervals for intermediate and unfavorable behavior scores in some subgroups? This raises questions about the reliability of the risk differentiation.
7. The manuscript lacks a thorough discussion of the biological mechanisms underlying the observed associations. How do high parity and early sexual initiation mechanistically increase cervical cancer risk? The biological pathways by which high parity increases risk (e.g., hormonal changes during pregnancy leading to cervical ectopy). How might PRS interact with behavioral risk factors at a molecular or physiological level?
Comments on the Quality of English LanguageNumerous grammatical errors and awkward phrases detract from the manuscript's clarity. Examples include “risk of sexual & reproductive behavior and PRS” and “early marriage challenges.” A professional language review is needed.
Author Response
First of all, we would like to express our sincere thanks to your helpful comments and suggestions on our manuscript, your time and efforts in handling our manuscript are much appreciated. The explanation of the modifications as well as corrections in this revision are listed as follow. The changes that we made have been marked with yellow in the revised manuscript.
Comments 1: The construction of the PRS is inadequately described. Simply stating the inclusion of 21 genetic variants is insufficient. Include detailed information on the genetic variants selected, the rationale behind their inclusion, and the validation of the PRS. The manuscript does not address potential confounding factors, such as population stratification or genetic ancestry differences, which may bias the PRS outcomes. Adjustments for these factors should be explicitly stated. The biological relevance of the selected variants. Are these variants causally linked to cervical carcinoma, or are they simply associated?
Response 1:
We appreciate your comment regarding the construction of the PRS. Detailed information about the PRS construction is provided in the supplementary materials (Page 3, lines 11–22). As described in the supplementary materials:
“Polygenic risk score construction for cervical carcinoma. We constructed the cervical carcinoma PRS for individuals in UKB based on the GWAS summary statistics in the FinnGen study3 (data freeze 8, Fall 2021), which included 2,913 cases and 149,394 controls. The clumping and threshold approach was applied to construct the cervical carcinoma PRS by selecting the significant (P<5×10−8) and independent SNPs (r2<0.1) in 1,000 kb region based on genotypes of 1,000 randomly selected European samples from UKB. The effect sizes of 21 significant and independent SNPs were derived from the FinnGen study (Table S1). We summed the effect of all selected SNPs together into a PRS as the following model,
PRSi =∑βkXik
where βk was the effect size of the variant k in the FinnGen study, and Xik was the number of effective alleles for SNP k (Xik=0, 1, or 2) of ith participant in UKB”. (“Supplementary materials”, Page 3, line 11-22)
Thank you for highlighting the importance of addressing confounding factors. As described in the supplementary materials: To minimize population stratification bias, we restricted our analysis to a European ancestry subset of 408,812 individuals, including those who self-identified as white British and demonstrated similar genetic ancestry based on principal component analysis (“Supplementary materials”, Page 3, line 3-10). These steps were implemented to control confounding factors such as population stratification.
We acknowledge the your concern regarding the biological relevance of the selected SNPs, and these SNPs are robustly associated with cervical carcinoma, establishing biological relevance of these SNPs would require additional functional studies.
Comments 2: The matching process based on age and smoking status is reasonable, but more justification is needed for these choices over other potentially confounding variables, such as socioeconomic status or HPV vaccination history.
Response 2:
Thank you for your insightful comment. We agree that controlling for other potential confounders, such as socioeconomic status, in the revised manuscript, we included the Townsend Deprivation Index as a covariate in our analysis to account for potential socioeconomic disparities within the UK Biobank cohort.
Regarding HPV vaccination history, we acknowledge its relevance; however, the UK Biobank cohort comprises individuals aged 40–69 years at recruitment, and HPV vaccination programs were only introduced in more recent years. Moreover, only a very small proportion of the cohort has vaccination data available, making it insufficient for robust statistical analysis.
Comments 3: Oral contraceptive use is included as an adjustment variable, but its potential as an independent modifier of risk is ignored. This oversight undermines the interpretability of the results.
Response 3:
Thank you for raising this point. We included oral contraceptive use as an adjustment variable in our analysis to account for its potential confounding effect. In our dataset of 7,049 individuals from the UK Biobank cohort, 6,374 participants reported using oral contraceptives, while only 657 participants had never used them. This substantial imbalance limits the statistical power to explore its role as an independent modifier. Given that, we focused on its inclusion as a covariate in the model rather than as an independent modifier. We agree that future studies with more detailed data on contraceptive use (such as duration, frequency of use, age at first use) could better address this question.
Comments 4: Interaction effects between PRS and behavioral scores are only briefly mentioned but not rigorously tested. A detailed assessment using interaction terms or stratified analyses is necessary to validate the combined risk model.
Response 4:
Thank you for your comment. In Table 8, we assessed the additive interactions between sexual & reproductive behavior and genetic risk on the risk of cervical carcinoma, which showed no significant interaction between them.
Comments 5: Why does high PRS contribute relatively modestly (OR 1.58) to risk compared to behavioral factors (e.g., OR 3.13 for multiple partners)? This requires deeper exploration of the biological underpinnings.
Response 5: Thank you for your comment. Our results indicate no interaction between sexual and reproductive behavior and genetic risk, suggesting that these factors independently contribute to cervical cancer risk. This means that unfavorable sexual and reproductive behaviors increase the risk regardless of an individual’s genetic predisposition. While the high PRS contributes modestly to risk, the stronger association with behavioral factors highlights the significant impact of modifiable lifestyle factors. This finding emphasizes the importance of maintaining healthy sexual and reproductive behaviors, even for individuals with low genetic risk, as behavioral interventions are more actionable and can have a substantial influence on reducing disease risk.
Comments 6: How do the authors reconcile the overlapping confidence intervals for intermediate and unfavorable behavior scores in some subgroups? This raises questions about the reliability of the risk differentiation.
Response 6: Thank you for your comment. While the confidence intervals for intermediate and unfavorable behavior scores overlap in some subgroups, the overall trend consistently indicates an increased risk from the intermediate to the unfavorable behavior groups. This suggests a gradient effect, where increasingly unfavorable behaviors are associated with higher risk. We believe this trend, despite the overlapping confidence intervals, supports the reliability of the risk differentiation and emphasizes the importance of reducing unfavorable behaviors to decrease cancer risk.
Comments 7: The manuscript lacks a thorough discussion of the biological mechanisms underlying the observed associations. How do high parity and early sexual initiation mechanistically increase cervical cancer risk? The biological pathways by which high parity increases risk (e.g., hormonal changes during pregnancy leading to cervical ectopy). How might PRS interact with behavioral risk factors at a molecular or physiological level?
Response 7: Thank you for your comment, we added “Younger female may have an immature cervix that is more susceptible to HPV infection, and early sexual debut often correlates with a higher number of sexual partners, increasing the likelihood of exposure to high-risk HPV strains. The biological vulnerability of the cervix in early sexual initiation may also be compounded by an underdeveloped immune system, leading to inadequate clearance of HPV infections” in discussion part. (“Discussion”, Page 12, lines 238–243).
Regarding your suggestion on how PRS might interact with behavioral risk factors at the molecular or physiological level, we agree that this is a crucial area for future exploration. At present, there is limited understanding of how genetic risk, as quantified by polygentic risk scores (PRS), interacts with environmental and behavioral factors, such as sexual and reproductive behaviors. We believe that further research will be needed to uncover the molecular interactions between SNPs/genetic risk and these behavioral factors, particularly how they might jointly influence lifestyle or cellular processes that contribute to the development of cervical carcinoma.
Round 2
Reviewer 2 Report
Comments and Suggestions for Authors
All comments previously raised were well addressed.